# Effectiveness of Antenatal Corticosteroids in reducing morbidities and mortality in Preterm neonates: Evidence from a Tertiary Level Hospital in Nepal

Prajwal Paudel[1]*, Shree Prasad Adhikari[1], Kalpana Upadhyaya Subedi[1], Shailendra Bir Karmacharya[1], Sandesh Poudel[1], Megha Mishra[1], Laxmiswori Prajapati[1], Lov Sah[2], Needa Shrestha[1], Subash Bhattarai[3], Pratiksha Bhattarai[4], Avinash K. Sunny[5]

1 Paropakar Maternity and Women's Hospital, Kathmandu, Nepal, 2 Madhesh Institute of Health Science, Janakpur, Nepal, 3 Noble Medical College and Teaching Hospital, Kathmandu University, Kathmandu, Nepal, 4 Noble College, Pokhara University, Pokhara, Nepal, 5 Nepalese Society of Community Medicine, Kathmandu, Nepal

* prajwal.paudel999@gmail.com

## Abstract

### Background

The use of antenatal corticosteroids (ACS) in mothers less than 34 weeks' period of gestation has shown promising results with significant reduction in neonatal mortality and morbidities in high income settings. This study was carried out to assess the effectiveness of ACS in terms of neonatal outcome in less than 34 weeks in resource limited settings.

### Methods

A prospective study was conducted from 15 March 2022 to 14 March 2023 among the babies born before 34 weeks' period of gestation (POG), in Paropakar maternity hospital, Nepal. Descriptive statistics using frequency and percentages was used to describe the socio-demographic, obstetric and neonatal characteristics. Multi-variable logistic regression analysis was done to assess the significance of ACS against various neonatal conditions.

### Results

Out of 358 preterm neonates (<34 weeks), 206 were born to mothers who received ACS and 152 to mothers who did not. Mothers having any complications during delivery were more likely to receive ACS, (69.7% vs 50.0%, p = 0.002). Newborns of mothers who received ACS had significantly lower rates of respiratory distress syndrome (21.8% vs 61.8%, p < 0.001), necrotizing enterocolitis (5.8% vs 19.7%, p < 0.001), perinatal asphyxia (18.4% vs 35.5%, p < 0.001), neonatal sepsis (32.0% vs

**Data availability statement:** All relevant data are within the manuscript and its Supporting Information files.

**Funding:** The author(s) received no specific funding for this work.

**Competing interests:** No authors have competing interests.

43.4%, p < 0.027), and need for mechanical ventilation (15.5% vs 41.4%, p < 0.001). Newborn of mothers who did not receive ACS had higher odds of respiratory distress syndrome (adjusted odds ratio (a0R): 4.181, 95% CI: 2.462–7.100) and the need for mechanical ventilation (a0R: 2.266, 95% CI: 1.300–3.950). Lack of exposure to ACS was associated with higher odds of prolonged hospital stay (aOR: 3.321, 95% CI: 1.957–5.638) and mortality (aOR: 5.731, 95% CI: 3.199–10.266).

## Conclusion

ACS was more frequently used in mothers of less than 34 weeks POG having some complications during pregnancy. Use of ACS in deliveries of less than 34 weeks POG was associated with reduced risk of RDS, NEC and need for Mechanical Ventilation along with decrease hospital stay and neonatal mortality. Strengthening national guidelines with recommendation for the use of ACS in mothers less than 34 weeks POG can avert deaths due to complications of prematurity and help save more newborns.

## Introduction

Worldwide, an approximate 15 million babies are born preterm (< 37 weeks' period of gestation) and are at a greater risk of mortality due to prematurity related complications [1]. Prematurity, which accounts for 35% of death in babies less than 28 days of life is the leading cause of neonatal mortality and a global challenge to achieve the Sustainable Development Goal (SDG) target of reducing neonatal mortality rate particularly in the Low- and Middle-income countries (LMIC) [2]. Developmental immaturity of various systems leading to Respiratory distress syndrome, Necrotizing enterocolitis, Bleeding problems including intraventricular hemorrhage are the worrisome complications leading to augmented risk of mortality in babies who are born preterm [3]. Of the preventive approaches tailored towards reducing preterm related complications after birth, administration of antenatal corticosteroids (ACS) has shown promising results in many High-income countries (HIC) [4].

Concerted efforts to improve antenatal care, management of high-risk pregnancies, provision of comprehensive obstetric care are the pillars to alleviate the burden related to prematurity and curb the mortality stemming out of it [5,6]. While various trials have assessed the vitality of antenatal corticosteroids in reduction of neonatal morbidity, stillbirth and neonatal mortality and being extensively used in high income settings, its use in LMICs is limited due to various constraints [7]. Adoption of Antenatal corticosteroids use in routine practice in preterm deliveries is a challenge possibly due to lack of national guidelines, prescribing authority, lack of orientation to healthcare workers, and lack of timely availability of drugs [7,8].

The World Health Organization (WHO) current guideline on preterm birth management suggest the use of a single course of ACS (dexamethasone or betamethasone,

24 mg administered by intramuscular injection in divided doses) to mothers less than 34 weeks' period of gestation (POG), [9]. Safe and effective use of ACS has led to improved birth outcomes and reduction in neonatal morbidity and mortality in HICs [4,10]. However, coverage of ACS in preterm deliveries remains low in LMICs, despite the fact that 99% of neonatal deaths occur in these settings [11]. While there are evidences that effective use of ACS has the potential to save over 2 million newborns annually, use of ACS as a part of obstetric and preterm deliveries management has not been successfully implemented [12].

In Nepal, newborn mortality rate has been stagnant for more than a decade, and prematurity related complications contribute to the major fraction of neonatal deaths [13]. Quality antenatal and Intrapartum care has always been an issue and comprehensive sick newborn care is lacking in most of the health facilities in Nepal [14]. Despite the initiation of free newborn care services, care for newborns who require mechanical ventilation, surfactant therapy and neonatal surgery is a unfinished agenda due to high out of pocket expenses for parents and lack of trained manpower and proper guideline and strategies of the government to address such issues [15,16]. In this context, strengthening of ACS use in deliveries less than 34 weeks can be a game changer to avert deaths due to prematurity related medical and surgical complications. In our study, we aimed to examine the effect of using ACS in preterm deliveries of less than 34 weeks POG on neonatal morbidities and mortality.

## Materials and methods

### Study design and setting

The study was conducted in Paropakar Maternity and Women's Hospital which serves 22,000–24,000 deliveries per year with normal deliveries conducted in labor rooms and complicated deliveries via caesarean section in operation theatres. The hospital provides level II and III newborn care to the admitted sick newborns in NICU (Neonatal intensive care unit), SNCU (Special newborn care unit) and KMC (Kangaroo mother care) unit. Despite the lack of national protocol on use of antenatal corticosteroids in deliveries less than 34 weeks, all the mothers delivering or at risk of delivery before 34 weeks POG are given ACS. The ACS currently being given is intramuscular dexamethasone and given to the mothers at the time of admission. However, for various reasons all of them don't receive the ACS. This was an observational study and no random assignment of ACS was done; all decisions were made by treating physicians.

### Data collection and management

All the babies born before 34 weeks of gestation are admitted in the newborn care unit along with recording of all the related antepartum, intrapartum condition of the mothers of the admitted preterm babies. Similarly, information regarding diagnosis, treatment provided, investigations done and outcome of the baby till discharge are recorded by the nursing staff or medical officers. All the data about mothers and newborns were collected by the medical doctors in forms designed to gather the relevant information. The forms that were completed were then assessed by the senior doctors for completeness. The cleaned data were exported into Statistical Package for the Social Sciences (SPSS) for further data analysis.

### Statistical analysis

Descriptive statistics using frequency and percentages were used to describe the socio-demographic, obstetric and neonatal characteristics. Binary logistic regression was performed to analyze the level of association between the characteristics and ACS use in mothers. The significance was determined at $p < 0.05$. All the variables with $p < 0.2$ in the univariate analysis were considered for multi-variable logistic regression analysis.

## Variables used in the study

Sick babies were classified as having any of the following morbidity [17]:

Complications of prematurity: Conditions like respiratory distress syndrome, necrotizing enterocolitis, apnea of prematurity, hypoglycemia and hypothermia.

Respiratory distress syndrome: a condition arising due to lack of surfactant in babies born prematurely and presenting with features of respiratory distress at or within 6 hours of life

Necrotizing enterocolitis: acute inflammatory condition of gastrointestinal tract in premature babies leading to abdominal distention and perforation and hemorrhage if not treated timely

Perinatal asphyxia: Apgar score < 3 at 1 min or < 7 at 5 minutes of birth, with clinical evidence or abnormal ABG (Arterial blood gas analysis).

Neonatal sepsis: Clinical signs of severe bacterial infection, with a blood culture positive for a pathogenic organism.

Neonatal jaundice: Babies with total Serum Bilirubin (TSB) increasing by > 5 mg/ dl/ day or 0.5 mg/ dl/ h, TSB > 15 mg/ dl, conjugated serum bilirubin > 2 mg/dl.

Meconium aspiration syndrome (MAS): Breathing problems that a newborn baby may have when there are no other causes, and the baby has passed meconium (stool) into the amniotic fluid during labor.

Low birth weight: Birth weight of the baby less than 2500 grams Preterm: Babies born before 37 weeks of gestation

## Ethical consideration

All the parents provided informed written consent before the start of the data collection and confidentiality was maintained. Ethical approval was received from Ethical Review Board of PMWH (reference number- 61/1808).

## Results

Out of total 23,915 births during the study period, 406 were still births and were excluded from the study. Among the remaining 23,509 live births, 1,860 were preterm, of which 358 were less than 34 weeks POG and 206 of these babies were born to mothers who received antenatal corticosteroids (ACS) whereas, 152 of the babies were born to the mothers who did not receive ACS. (Fig 1)

Table 1 shows the bivariate association of characteristics among mothers with less than 34 weeks POG who received (n = 206) and did not receive (n = 152) antenatal corticosteroids (ACS). There were no significant differences in maternal age groups (p = 0.742), ethnicity (p = 0.491), parity (p = 0.530) and ANC visits (p = 0.156) between mothers who received and those who did not receive ACS. However, the proportion of mothers who experienced complications during delivery were significantly higher among those who received ACS compared to those who did not (69.7% vs 50.0%; p = 0.002).

Table 2 compares the morbidities among newborns whose mothers received and did not receive antenatal corticosteroids (ACS). Newborns of mothers who received ACS had significantly lower rates of respiratory distress syndrome (21.8% vs 61.8%, p < 0.001), necrotizing enterocolitis (5.8% vs 19.7%, p < 0.001), perinatal asphyxia (18.4% vs 35.5%, p < 0.001), neonatal sepsis (32.0% vs 43.4%, p < 0.027), and requirement for mechanical ventilation (15.5% vs 41.4%, p < 0.001) compared to newborns whose mothers did not receive ACS. There were no significant differences in the rates of neonatal jaundice, meconium aspiration syndrome, hypoglycemia or apnea between the two groups.

Table 3 provides the multivariate analysis of morbidities among newborns whose mothers received or did not receive ACS. After adjusting for potential confounders, newborns whose mothers did not receive ACS were four times more likely to develop respiratory distress syndrome (adjusted odds ratio (a0R): 4.181, 95% CI: 2.462–7.100) and two times more likely to require mechanical ventilation (a0R: 2.266, 95% CI: 1.300–3.950) compared to those whose mothers received ACS.

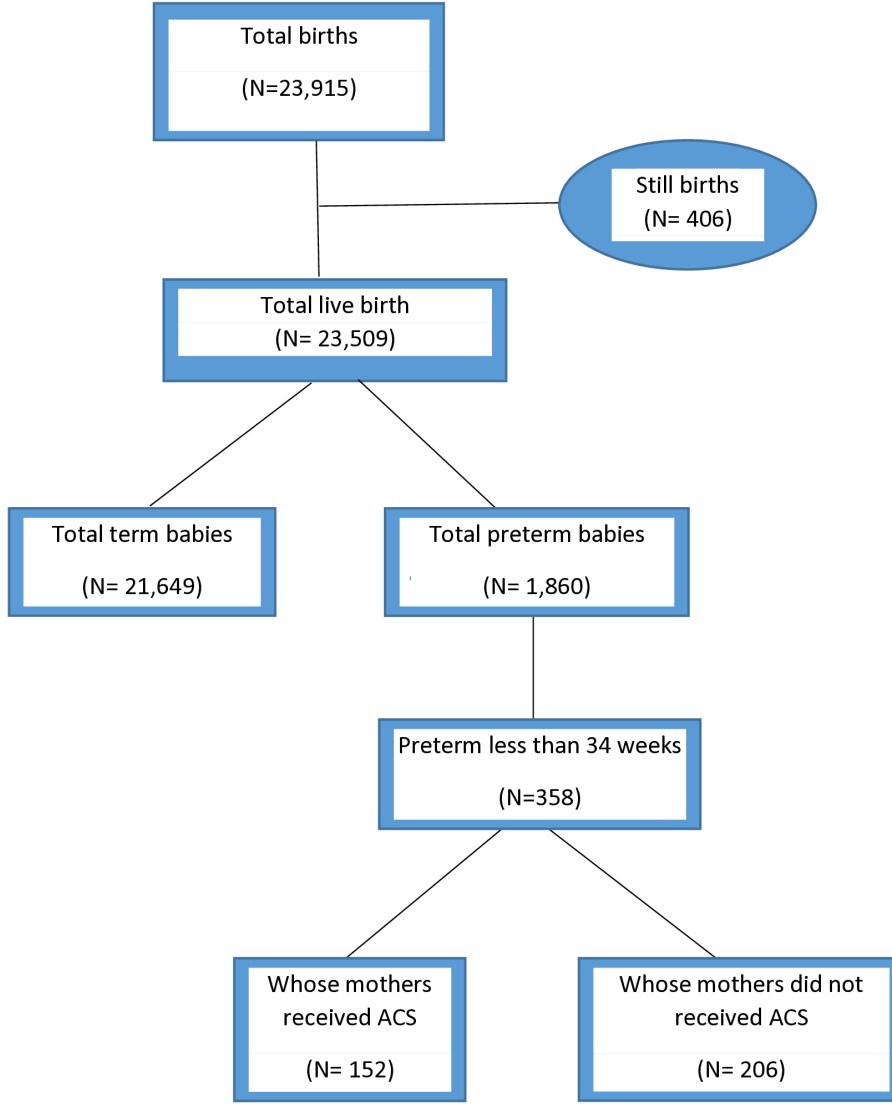

**Fig 1. Flow diagram of participant selection.**

Table 4 presents the multivariate analysis of outcomes among newborns whose mothers received or did not receive ACS. Newborns whose mothers did not receive ACS were five times more likely to result in mortality (aOR: 5.731, 95% CI: 3.199–10.266) and prolonged duration of stay in the hospital (aOR: 3.321, 95% CI: 1.957–5.638) compared to newborns whose mothers received ACS. There was no significant difference for receiving KMC (aOR: 1.131, 95% CI: 0.652–1.961) and ACS administration.

## Discussion

This study sheds light on the usefulness of administering ACS to alleviate the burden of complications of prematurity like RDS, NEC and mortality among these babies. This is one of the rare studies done in our context because not all hospitals have institutionalized the use of ACS due to lack of national protocol and recommendation to use it. During the study

**Table 1. Bivariate association of characteristics of mothers (<34 weeks POG) who received and did not receive ACS.**

| Indicator | Antenatal Corticosteroid (ACS) | | Total | p-value |
|---|---|---|---|---|
| **Maternal Age*** | No (152) | Yes (206) | | 0.742 |
| <20 | 10(7.6%) | 16(9.4%) | 26(8.4%) | |
| 20-35 | 107(81.1%) | 144(81.8%) | 251(81.5%) | |
| >35 | 15(11.4%) | 16(9.1%) | 31(10.1%) | |
| **Ethnicity** | | | | 0.491 |
| Dalit | 21(13.8%) | 34(16.5%) | 55(15.4%) | |
| Janajati | 65(42.8%) | 85(41.3%) | 150(41.9%) | |
| Madhesi | 8(5.3%) | 10(4.9%) | 18(5.0%) | |
| Muslim | 4(2.6%) | 1(0.50%) | 5(1.40%) | |
| Brahmin/Chhetri | 54(35.5%) | 76(36.9%) | 130(36.3%) | |
| **Parity*** | | | | 0.530 |
| Nullipara | 49(40.2%) | 52(35.9%) | 101(37.8%) | |
| Primipara | 51(41.8%) | 59(40.7%) | 110(41.2%) | |
| Multipara | 22(18.0%) | 34(23.4%) | 56(21.0%) | |
| **Complications during delivery*** | | | | 0.002 |
| No | 50(50.0%) | 44(30.3%) | 94(38.4%) | |
| Yes | 50(50.0%) | 101(69.7%) | 151(61.6%) | |
| **ANC Visits** | | | | 0.156 |
| No Visits | 29(19.1%) | 36(17.5%) | 65(18.2%) | |
| <4 Visit | 27(17.8%) | 23(11.2%) | 50(14.0%) | |
| ≥4 Visit | 96(63.2%) | 147(71.4%) | 243(67.9%) | |

*Missing data

period, a total of 358 preterm newborns were admitted in NICU <34 weeks and of them 206 were born to mothers who received ACS.

Our study showed more than four-fold risk of RDS in premature babies <34 weeks born to mother not receiving ACS which is in line with the other studies which found a lower frequency of RDS among babies whose mothers were antenatally treated with steroids [18,19]. To the contrary, results vary in studies examining the effects of ACS on fetal lung maturity showing no significant risk of RDS in preterm babies [20,21]. ACS exerts beneficial effects on lung maturation and respiratory function by enhancing tissue and alveolar surfactant production, promoting lung volume and parenchymal maturation while decreasing vascular permeability [22]. In some studies, RDS was not significantly prevented with the use of ACS probably due to other complications the premature babies had or might be due to compromised quality care.

This study depicted that administration of antenatal corticosteroid could reduce preterm- related deaths by 5 folds compared to the premature babies whose mothers were not provided any ACS. Several studies have concluded similar findings with reduced preterm neonatal mortality rates when ACS was administered to the mothers [4,23–25]. The major causes of death in Preterm babies are due to the developmental immaturity of the respiratory, gastrointestinal, along with other systems which lead to complications like RDS, NEC, hematological disorders.

Length of hospital stay in newborns born to mothers not receiving ACS was more than three-folds higher as compared to the newborns born to mothers receiving ACS. The use of ACS exerts a beneficial role in prevention of complications arising due to prematurity. Conditions like NEC, RDS, bleeding disorders and probably decreased morbidities have shown a decline in hospital stay such cohort.

**Table 2. Bivariate analysis of morbidities among newborns whose mothers received or did not receive ACS.**

| Indicator | Antenatal Corticosteroid (ACS) | | p-value |
|---|---|---|---|
| | No | Yes | |
| **Respiratory Distress Syndrome** | | | |
| No | 58(38.2%) | 161(78.2%) | <0.001 |
| Yes | 94(61.8%) | 45(21.8%) | |
| **NEC** | | | <0.001 |
| No | 122(80.3%) | 194(94.2%) | |
| Yes | 30(19.7%) | 12(5.8%) | |
| **Perinatal Asphyxia** | | | <0.001 |
| No | 98(65.4%) | 168(81.6%) | |
| Yes | 54(35.5%) | 38(18.4%) | |
| **Neonatal Sepsis** | | | 0.027 |
| No | 86(56.6%) | 140(68.0%) | |
| Yes | 66(43.4%) | 66(32.0%) | |
| **Neonatal Jaundice** | | | 0.320 |
| No | 130(85.5%) | 168(81.6%) | |
| Yes | 22(14.5%) | 38(18.4%) | |
| **Meconium Aspiration Syndrome (MAS)** | | | 0.185 |
| No | 149(98.0%) | 205(99.5%) | |
| Yes | 3(2.0%) | 1(0.5%) | |
| **Hypoglycemia** | | | 0.441 |
| No | 144(94.7%) | 191(92.7%) | |
| Yes | 8(5.3%) | 15(7.3%) | |
| **Mechanical Ventilation** | | | <0.001 |
| No | 89(58.6%) | 174(84.5%) | |
| Yes | 63(41.4%) | 32(15.5%) | |
| **Apnea** | | | 0.797 |
| No | 142(93.4%) | 191(92.7%) | |
| Yes | 10(6.6%) | 15(7.3%) | |

Of the babies whose mothers received ACS compared to who did not, the occurrence of NEC, Perinatal Asphyxia, and the need for mechanical ventilation was lower in the study. Decreased incidence of NEC has also been seen in previous studies [18,26]. ACS administration promotes the maturation of the intestinal mucosal barrier through decrement in bacteria translocation, reduction in uptake of macromolecules, and lowering intestinal permeability [27]. Interestingly, those babies lacked association with the incidence of Perinatal asphyxia compared to babies born to mothers who did not receive ACS. Studies done earlier have also shown similar effects in such cohort of babies [18,26].

## Conclusion

Though the use of ACS is greatly underused in Low-income settings, our prospective study done to estimate the effects of ACS in preterm deliveries less than 34 weeks POG, established an association among babies born to mothers receiving ACS and occurrence of RDS, NEC perinatal asphyxia and need for mechanical ventilation in our settings. Administration of ACS was proven to be a high impact intervention according to our study with a potential to reduce preterm deaths by five folds. Larger studies are warranted to determine the exact impact of ACS on neonatal morbidities and mortality.

**Table 3. Multivariate analysis of the morbidities among the newborns who mothers receive or did not receive ACS.**

| Indicator | aOR (95%CI) | p-value |
|---|---|---|
| **Respiratory Distress Syndrome** | | <0.0001 |
| No | Ref | |
| Yes | 4.181 (2.462-7.100) | |
| **NEC** | | 0.552 |
| No | Ref | |
| Yes | 1.279 (0.569-2.878) | |
| **Perinatal Asphyxia** | | 0.137 |
| No | Ref | |
| Yes | 1.513 (0.877-2.609) | |
| **Neonatal Sepsis** | | 0.534 |
| No | Ref | |
| Yes | 1.170 (0.714-1.917) | |
| **Meconium Aspiration Syndrome (MAS)** | | 0.154 |
| No | Ref | |
| Yes | 1.717 (0.522-2.658) | |
| **Mechanical Ventilation** | | 0.004 |
| No | Ref | |
| Yes | 2.266 (1.300-3.950) | |

**Table 4. Multivariate analysis of the outcomes among the newborns who mothers receive or did not receive ACS.**

| Outcome | Antenatal Corticosteroid (ACS) | | P-value | aOR(95%CI) | P-value |
|---|---|---|---|---|---|
| | **No** | **Yes** | | | |
| **Mortality** | | | <0.001 | | <0.001 |
| No | 85(55.9%) | 181(87.9%) | | Ref | |
| Yes | 67(44.1%) | 25(12.1%) | | 5.731 (3.199-10.266) | |
| **KMC received** | | | 0.001 | | 0.182 |
| Yes | 31(20.4%) | 75(36.4%) | | Ref | |
| No | 121(79.6%) | 131(63.6%) | | 1.131 (0.652-1.961) | |
| **Duration of stay** | | | <0.001 | | <0.001 |
| <14 days | 94(61.8%) | 172(83.5%) | | Ref | |
| >14 days | 58(38.2%) | 34(16.5%) | | 3.321 (1.957-5.638) | |

## Supporting information

**S1 Data.  Supplementary file ACS.**
(XLSX)

## Acknowledgments

We would like to acknowledge all the participants involved in this study and the staff of Paropakar Maternity and Women's Hospital.

## Author contributions

**Conceptualization:** Prajwal Paudel, Pratiksha Bhattarai.

**Data curation:** Sandesh Poudel, Needa Shrestha, Pratiksha Bhattarai, Avinash K Sunny.

**Formal analysis:** Pratiksha Bhattarai, Avinash K Sunny.

**Investigation:** Prajwal Paudel, Sandesh Poudel, Megha Mishra, Subash Bhattarai.

**Methodology:** Prajwal Paudel, Megha Mishra, Subash Bhattarai.

**Resources:** Kalpana Upadhyaya Subedi, Laxmiswori Prajapati.

**Software:** Laxmiswori Prajapati, Needa Shrestha, Avinash K Sunny.

**Supervision:** Prajwal Paudel, Shree Prasad Adhikari.

**Validation:** Avinash K Sunny.

**Visualization:** Avinash K Sunny.

**Writing – original draft:** Prajwal Paudel.

**Writing – review & editing:** Prajwal Paudel, Shailendra Bir Karmacharya, Lov Sah, Subash Bhattarai, Pratiksha Bhattarai, Avinash K Sunny.

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
