## [Decision Letter · Decision Letter 0]

30 Jun 2025

Dear Dr. Paudel,

Thank you for submitting your manuscript to PLOS ONE. After careful consideration, we feel that it has merit but does not fully meet PLOS ONE’s publication criteria as it currently stands. Therefore, we invite you to submit a revised version of the manuscript that addresses the points raised during the review process.

We look forward to receiving your revised manuscript.

Kind regards,

Vinay Shukla, M.Sc., Ph.D.

Guest Editor

PLOS ONE

Journal Requirements:

2. Please provide additional details regarding participant consent. In the ethics statement in the Methods and online submission information, please ensure that you have specified (1) whether consent was informed and (2) what type you obtained (for instance, written or verbal, and if verbal, how it was documented and witnessed).

No authors have competing interests.

5. Please amend the manuscript submission data (via Edit Submission) to include author Kalpana Upadhyaya Subedi.

6. Please amend your authorship list in your manuscript file to include author Kalpana Subedi.

7. Please amend your list of authors on the manuscript to ensure that each author is linked to an affiliation. Authors’ affiliations should reflect the institution where the work was done (if authors moved subsequently, you can also list the new affiliation stating “current affiliation:….” as necessary).

8. Please include a caption for figure 1.

9. Please ensure that you refer to Figure 1 in your text as, if accepted, production will need this reference to link the reader to the figure.

10. Please include captions for your Supporting Information files at the end of your manuscript, and update any in-text citations to match accordingly. Please see our Supporting Information guidelines for more information: http://journals.plos.org/plosone/s/supporting-information.

Additional Editor Comments:

The manuscript titled “Effectiveness of Antenatal Corticosteroids in Reducing Morbidities and Mortality in Preterm Neonates: Evidence from a Tertiary Level Hospital in Nepal” presents important findings on the use of antenatal corticosteroids (ACS) in a resource-limited setting. Reviewer 1 raised key concerns regarding clarity on the study design (randomization and ethical considerations), missing information on timing of steroid administration, and minor typographical errors in the results section. Reviewer 2 acknowledged the significance of the study and its potential impact but recommended improving the presentation of statistical data (e.g., consistent reporting of 95% CIs) and addressing possible confounders such as maternal subclinical infections. Addressing these methodological and editorial points will enhance the scientific quality and clarity of the manuscript.

Reviewers' comments:

Reviewer's Responses to Questions

**Comments to the Author**

1. Is the manuscript technically sound, and do the data support the conclusions?

Reviewer #1: Yes

Reviewer #2: Yes

2. Has the statistical analysis been performed appropriately and rigorously?

Reviewer #1: Yes

Reviewer #2: Yes

3. Have the authors made all data underlying the findings in their manuscript fully available?

Reviewer #1: Yes

Reviewer #2: Yes

4. Is the manuscript presented in an intelligible fashion and written in standard English?

Reviewer #1: Yes

Reviewer #2: Yes

Reviewer #1: 1-It is not clear how the two groups were chosen. Is this a randomised study? And if so, can this procedure be considered ethically correct?

2- The time between the course of corticosteroids and delivery was not given.

3- the 3rd paragraph of the results table 2 instead of table 1

4- the 3rd line of the results lacks a space between POG. and 206

Reviewer #2: The evidence of the effectiveness of antenatal steroids in high resource setting is undebatable, yet the data for their use in limited resource settings is limited. Here the authors have presented a compelling and needed study supporting the use of antenatal steroid use in a limited resource setting.

Introduction:

SDG target - please write out any acronyms in full on first use

POG - please write out any acronyms in full on first use

Results:

The authors state in the results sections that "distribution of outcomes for both groups, p-values indicating the statistical significance, and adjusted odds ratios (aOR) with confidence intervals (95% CI) for each outcome". When reporting the results in text, there is a lot of repetition to the tables. The authors also do not report 95% CIs in text, but just state 95% CI which makes the results very cluttered and difficult to read, e.g. "administration of ACS is significantly associated with the development of RDS 38.2% (58) vs 21.8% (45), (p-value=<0.0001) and (aOR= 4.181, 95% CI). I would suggest simply reporting percentages and p-values, or just p-values and reference tables reduce confusion and repitition. Otherwise please include the actual 95% CI values in text and not simply state "aOR= ..., 95% CI".

There was a greater number of preterm infants who did not receive ACS that had neonatal sepsis compared to preterm infants that received ACS. Is this due to a greater number developing infection after birth in non-ACS infants compared to preterm infants whose received ACS? Or were there potentially a greater number of non-ACS mothers having sub-clinical bacterial infections that they passed on to their infants, and this lead to sepsis postnatally? Similarly were there a higher number of mothers that had overt infection and received prenatal antibiotics prior to preterm birth that could have influenced this outcome? Subclinical infection in mothers can be a risk factor for preterm birth as well as RSD, could this be a confounder in the outcomes of the study?

**Do you want your identity to be public for this peer review?** For information about this choice, including consent withdrawal, please see our Privacy Policy

Reviewer #1: **Yes: ** Mahdi Farhati

Reviewer #2: **Yes: ** Angela Cumberland

---

## [Author Response · Author response to Decision Letter 1]

24 Sep 2025

Reviewer #1 comments

Comment 1-It is not clear how the two groups were chosen. Is this a randomised study? And if so, can this procedure be considered ethically correct?

Response: This is not a randomized study. We have specifically mentioned this in line 95 and 96 “This was an observational study and no random assignment of ACS was done; all decisions were made by treating physicians”.

Comment 2- The time between the course of corticosteroids and delivery was not given.

Response: We acknowledge this as a limitation of the study as we could not collect the data on the time interval.

Comment 3- the 3rd paragraph of the results table 2 instead of table 1

Response: We have now corrected this in the result section with table 2.

Comment 4- the 3rd line of the results lacks a space between POG. and 206

Response: We have now corrected this typographical error.

Reviewer #2:

Comment 1-The evidence of the effectiveness of antenatal steroids in high resource setting is undebatable, yet the data for their use in limited resource settings is limited. Here the authors have presented a compelling and needed study supporting the use of antenatal steroid use in a limited resource setting.

Response: Thank you for this thoughtful comment. We appreciate your acknowledgement of the importance and relevance of our study.

Comment 2-

Introduction:

SDG target - please write out any acronyms in full on first use

POG - please write out any acronyms in full on first use

Response: We have revised this as suggested

Comment 3-

Results:

The authors state in the results sections that "distribution of outcomes for both groups, p-values indicating the statistical significance, and adjusted odds ratios (aOR) with confidence intervals (95% CI) for each outcome". When reporting the results in text, there is a lot of repetition to the tables. The authors also do not report 95% CIs in text, but just state 95% CI which makes the results very cluttered and difficult to read, e.g. "administration of ACS is significantly associated with the development of RDS 38.2% (58) vs 21.8% (45), (p-value=<0.0001) and (aOR= 4.181, 95% CI). I would suggest simply reporting percentages and p-values, or just p-values and reference tables reduce confusion and repitition. Otherwise please include the actual 95% CI values in text and not simply state "aOR= ..., 95% CI".

Response: We have revised this clearly referring to the tables reducing confusion and repetition.

Comment 4-

There was a greater number of preterm infants who did not receive ACS that had neonatal sepsis compared to preterm infants that received ACS. Is this due to a greater number developing infection after birth in non-ACS infants compared to preterm infants whose received ACS? Or were there potentially a greater number of non-ACS mothers having sub-clinical bacterial infections that they passed on to their infants, and this lead to sepsis postnatally? Similarly were there a higher number of mothers that had overt infection and received prenatal antibiotics prior to preterm birth that could have influenced this outcome? Subclinical infection in mothers can be a risk factor for preterm birth as well as RSD, could this be a confounder in the outcomes of the study?

Response: We acknowledge this could be a potential confounder, which could be assessed as this data were not collected

---

## [Editor Report · Decision Letter 1]

26 Sep 2025

Effectiveness of Antenatal Corticosteroids in reducing morbidities and mortality in Preterm neonates: Evidence from a Tertiary Level Hospital in Nepal.

PONE-D-24-54032R1

Dear Dr. Paudel,

We’re pleased to inform you that your manuscript has been judged scientifically suitable for publication and will be formally accepted for publication once it meets all outstanding technical requirements.

Kind regards,

Vinay Shukla, Ph.D.

Guest Editor

PLOS ONE
---

## [Editor Report · Acceptance letter]

PONE-D-24-54032R1

PLOS ONE

Dear Dr. Paudel,

I'm pleased to inform you that your manuscript has been deemed suitable for publication in PLOS ONE. Congratulations! Your manuscript is now being handed over to our production team.

Kind regards,

on behalf of

Dr. Vinay Shukla

Guest Editor

PLOS ONE